# Clinical Trends in Management of Locally Advanced ESCC: Real-World Evidence from a Large Single-Center Cohort Study

**DOI:** 10.3390/cancers14194953

**Published:** 2022-10-10

**Authors:** Yeong Jeong Jeon, Junsang Yoo, Jong Ho Cho, Young Mog Shim

**Affiliations:** 1Department of Thoracic and Cardiovascular Surgery, Samsung Medical Center, Sungkyunkwan University School of Medicine, Seoul 06351, Korea; 2Department of Digital Health, SAIHST, Sungkyunkwan University, Seoul 06355, Korea; 3Patient-Centered Outcomes Research Institute, Samsung Medical Center, Seoul 06355, Korea

**Keywords:** esophageal cancer, treatment, trend

## Abstract

**Simple Summary:**

Neoadjuvant chemoradiation followed by surgery (NCRT+S) has been widely applied to patients with locally advanced esophageal squamous cell carcinoma (ESCC); however, treatment trends and their survival outcomes in a real-world clinical setting are poorly understood. This study analyzed real-world evidence to understand treatment patterns and outcomes for 2151 patients with locally advanced ESCC by synthesizing the individuals’ general characteristics, cancer information, and treatment records extracted from the Clinical Data Warehouse from 1994 to 2018. Patients with NCRT+S had the most favorable 5-year overall survival (5yOS) with 58.1% (53–63.7%), although that for patients with upfront surgery was 48.6% (45.9–51.5%, *p* < 0.001). Moreover, patients who received adjuvant therapy after surgery had better 5yOS than those with surgery alone (58.4% (52.7–64.7%) vs. 47.3% (44.1–50.7%), *p* < 0.001). In conclusion, NCRT+S was the most effective treatment for locally advanced ESCC, and adjuvant chemotherapy may be an encouraging therapeutic option for patients with positive nodes after upfront surgery.

**Abstract:**

Neoadjuvant chemoradiation followed by surgery (NCRT+S) has been widely applied to patients with locally advanced esophageal squamous cell carcinoma (ESCC); however, treatment trends and their survival outcomes in a real-world clinical setting are poorly understood. This study aimed to analyze real-world evidence to understand treatment patterns and outcomes for patients with ESCC. We analyzed the treatment pattern and 5-year overall survival (5yOS) by synthesizing the individuals’ general characteristics, cancer information, and treatment records extracted from the Clinical Data Warehouse from 1994 to 2018. Of a total of 2151 patients, most patients received upfront surgery and 5yOS was 36.8% (31.4–43.1%). From 2003 to 2012, the use of NCRT increased, and 5yOS was improved to 42.2% (38.8–45.7%). Notably, after 2013, the proportion of NCRT+S markedly increased up to >50% of patients: 5yOS was much improved to 56.3% (53.2–59.6%). With regard to treatment, patients with NCRT+S had the most favorable 5yOS of 58.1% (53–63.7%), although that for patients with upfront surgery was 48.6% (45.9–51.5%, *p* < 0.001). Moreover, patients who received adjuvant therapy after surgery had better OS than those with surgery alone (58.4% (52.7–64.7%) vs. 47.3% (44.1–50.7%), *p* < 0.001). This analysis of real-world data demonstrated a significantly improved survival outcome for locally advanced ESCC over time since NCRT prior to surgery had been routinely applied. We revealed that NCRT+S was the most effective treatment for locally advanced ESCC and that adjuvant chemotherapy may be an encouraging therapeutic option for patients with positive nodes after upfront surgery.

## 1. Introduction

Esophageal cancer is the eighth most common malignant tumor worldwide and has a poor prognosis. For decades, multimodal treatment strategies for locally advanced esophageal squamous cell carcinoma (ESCC) have been studied for their benefits to improve survival. Randomized trials have shown that neoadjuvant chemoradiotherapy (CRT) or chemotherapy before surgery significantly improves survival for locally advanced ESCC [1,2,3,4]. As the experience with trimodality therapy has grown, treatment patterns for patients with locally advanced ESCC have changed; however, treatment trends and their survival outcomes in a real-world clinical setting remain poorly understood.

The term real-world evidence (RWE) describes data on the use and outcomes of different therapies in the real world of clinical practice. This type of evidence may include information from single-institution cohort studies, population-based studies, and nationwide data studies [5]. Such information may demonstrate the benefits of treatments in more generalized populations, whereas clinical trials may restrict enrollment to patients with a good performance status and may also include control patients. In addition, RWE can be used to investigate different treatment patterns within the real-world population over time. Therefore, RWE provides valuable information about the safety and effectiveness of different therapeutic strategies in clinical practice and explores trends in overall outcomes over time. This study aimed to analyze RWE to better understand treatment patterns and trends in outcomes for patients with locally advanced ESCC.

## 2. Patients and Methods

### 2.1. Study Setting

This study was conducted at an academic tertiary hospital with 1989 inpatient beds, an annual outpatient volume of 2.3 million, and an annual esophageal surgery volume of 200 cases. Patients were diagnosed by endoscopic biopsy, computed tomography (CT) of the chest or abdomen, and positron emission tomography (PET)-CT. The dataset used in this retrospective study was extracted from “DARWIN-C”, a clinical data warehouse (CDW). The institutional ethics committee approved this study (IRB No. 2021-04-122).

### 2.2. Eligibility Process

The inclusion criteria were as follows: age of 18 years and older and a diagnosis of esophageal ESCC between November 1994 and December 2018. The following were the exclusion criteria: (1) no administration of any anticancer treatment at the study site after diagnosis; (2) lack of availability of CDW information necessary for data analysis; and (3) failure to meet the diagnostic criteria for locally advanced esophageal cancer, defined as clinical T3/4 or N1–3 staging or as receipt of neoadjuvant CRT before surgery. 

### 2.3. Treatment

A multidisciplinary approach was used to determine the treatment for locally advanced ESCC. In cases of resectable tumors, patients underwent surgery with or without neoadjuvant therapy. Patients who had a resectable tumor without node metastasis underwent upfront surgery, and patients with node metastasis received neoadjuvant therapy before surgery. The neoadjuvant therapy consisted of concurrent external beam radiation and chemotherapy. Two cycles of intravenous chemotherapy were administered at 3-week intervals: 5-fluorouracil 1000  mg/m^2^/day for days 1–4, plus cisplatin 60  mg/m^2^/day on day 1. Radiation was delivered at 44 Gy in daily 2.0 Gy fractions over 4–5 weeks. Some individuals who were referred from other institutions underwent neoadjuvant chemotherapy without radiation. Tumor response and resectability were re-evaluated before surgical resection by chest CT and PET-CT within 3–4  weeks of completion of the neoadjuvant CRT. If the disease had not progressed, surgical treatment was performed.

For patients with a close margin after curative resection, adjuvant radiotherapy was applied. For these patients, 54–64 Gy of radiation was delivered in 27–32 fractions at 2.0 Gy per fraction. In addition, for patients with positive lymph nodes (pN+) after surgery, adjuvant chemotherapy was applied 4–8 weeks after surgery. The chemotherapy consisted of cisplatin (60 mg/m^2^, intravenously) and 5-fluorouracil (5-FU; 1000 mg/m^2^/day) in a continuous infusion for 4 days. Three to six cycles were administered at 3-week intervals. From 2005 to 2010, patients who were enrolled in a clinical trial received capecitabine (1000 mg/m^2^, twice a day, per oral, days 1–14) and cisplatin (75 mg/m^2^/day, intravenously, day 1). Each cycle was repeated every 3 weeks (four cycles) [6]. From 2011 to 2015, patients who were enrolled in another clinical trial received leucovorin and 5-FU (LV5FU2) or LV5FU2 plus oxaliplatin (FOLFOX) combination chemotherapies. The LV5FU2 regimen consisted of 2-week cycles of 200 mg/m^2^ leucovorin and a bolus injection of 5-FU (400 mg/m^2^, intravenously, day 1) followed by a 46 h continuous infusion of 5-FU (2400 mg/m^2^). The FOLFOX regimen consisted of 2-week cycles of oxaliplatin (85 mg/m^2^, intravenously, day 1) before administration of the LV5FU2 regimen. Other patients were observed without adjuvant chemotherapy because of the patients’ poor general condition or refusal or on the basis of the physician’s judgment. 

Patients who refused surgery after neoadjuvant CRT were followed without additional treatment or treated with additional radiation. Patients who had a poor performance status or unresectable tumors underwent definitive CCRT. When initially unresectable tumors became resectable after definitive CCRT, or esophageal rupture or fistula developed during definitive CCRT, esophagectomy was conducted. In the case of patients who had contraindications to radiation, palliative chemotherapy was applied. 

### 2.4. Data Analysis

We conducted a descriptive analysis of the treatment pattern, disease course, and survival outcomes by synthesizing the general characteristics, cancer data, and treatment records for individual patients. The baseline characteristics of the patients between the groups were compared using the ANOVA test for continuous variables, and the chi-square test or Fisher’s exact test was used for categorical variables, when appropriate.

The primary endpoint of the study was overall survival (OS). OS was calculated from the date of diagnosis of ESCC to the date of death from any cause or the last follow-up date. Based on the CDW annual data retrieval of death records from the National Statistics Korea (KOSTAT), patients without a death record until December 2020 were considered as alive until that time point. Survival analysis was used to investigate OS to compare and classify data in three sets according to the time period and the treatment received: before routine PET-CT assessment (1997–2002), after routine PET-CT and before NCRT administration (2003–2012), and after routine administration of NCRT for advanced ESCC (2013–2018). The comparison of OS among the periods and treatment groups was performed using the Kaplan–Meier method. We estimated the hazard ratio using univariate and multivariate Cox proportional-hazard regression model. Additionally, in order to adjust the TNM stage, we also employed a Cox proportional-hazard regression model stratified by stage and type of treatment.

## 3. Results

### 3.1. Study Population

We retrospectively reviewed the anonymized patient electronic medical records from a CDW of 5467 patients who were diagnosed with ESCC between November 1994 and December 2018. Patients who did not receive any treatment (*n* = 1402), lacked the necessary data for analysis (*n* = 757), or did not have locally advanced thoracic ESCC (*n* = 1157) were excluded. A total of 2151 patients were included in the final analysis (Figure 1).

### 3.2. Baseline Characteristics

The mean age was 63.1 ± 8.5 years, and most patients were male (94%). The Eastern Cooperative Oncology Group performance status was 0 for 1637 (76.1%), 1 for 180 (8.4%), and ≥2 for 57 (2.6%) patients. The tumor locations were cervical in 51 (2.4%), thoracic esophagus in 1939 (90.1%), and esophagogastric junction in 16 (0.7%) patients. Clinical characteristics of these patients are shown in Table 1. 

The number of patients treated for locally advanced ESCC increased over time. From 1994 to 2002, 261 patients were treated; most patients received upfront surgery. From 2003 to 2012, the number of patients treated increased to 790, and the number of multimodality treatments also increased after 2003. Notably, after 2013, 1100 patients were treated, and the proportion of treatment with NCRT greatly increased up to more than 50% of surgical cases for locally advanced ESCC. The practice patterns for each year are illustrated in Figure 2.

Of the target study population, 1299 (60.4%), 419 (19.5%), and 181 (8.41%) patients received upfront surgery, neoadjuvant CCRT followed by surgery (NCRT+S), and definitive CCRT on an intention-to-treat basis, respectively. Of the 1299 patients who received upfront surgery, 378 (29.1%) had adjuvant treatment: 283 chemotherapy, 90 radiotherapy, and 5 combined chemoradiotherapy. Of the 419 patients who received NCRT+S, 39 (9.3%) had adjuvant treatment: 38 chemotherapy and 1 radiotherapy. Figure 3 presents a Sankey diagram that shows the initial and sequential treatment patterns for patients with locally advanced ESCC.

### 3.3. Survival Outcomes

From 1994 to 2002, the 5-year OS of patients treated for locally advanced ESCC was 36.8% (95% confidence interval (CI), 31.4–43.1%). From 2003 to 2012, the 5-year OS rate was improved to 42.2% (95% CI, 38.8–45.7%). Notably, after 2013, when the use of NCRT for advanced ESCC became routine, the 5-year OS rate was much improved to 56.3% (95% CI, 53.2–59.6%). Kaplan–Meier survival curves for patients treated according to the period are shown in Figure 4.

According to the type of initial treatment, patients who received neoadjuvant therapy followed by surgery had more favorable outcomes. The 5-year OS for patients who underwent NCRT+S was 58.1% (53–63.7%), although that for patients with upfront surgery was 48.6% (45.9–51.5%; *p* < 0.001). A significantly reduced risk of death was observed for patients who received NCRT+S compared with that for patients who underwent upfront surgery alone (HR 0.74; 95% CI, 0.62–0.88; *p* < 0.001). Meanwhile, the 5-year OS of patients with definitive CCRT was 51.7%. Kaplan–Meier survival curves for patients according to the type of initial treatment are shown in Figure 5A. 

We also analyzed survival outcomes in patients who underwent treatment with curative intent considering the effect of adjuvant chemotherapy (Figure 5B). Patients who received adjuvant chemotherapy following upfront surgery had better 5-year OS than those treated with surgery alone (58.4% (52.7–64.7%) vs. 47.3% (44.1–50.7%)). Surgery with adjuvant chemotherapy significantly reduced the risk of death in comparison with surgery only (HR 0.69; 95% CI, 0.56–0.85; *p* < 0.001) Furthermore, patients who underwent upfront surgery with adjuvant chemotherapy had comparable outcomes to patients who underwent NCRT+S (58.4% (52.7–64.7%) vs. 58.1% (53–63.7%)). There was no difference in survival outcome between NCRT+S and surgery with adjuvant chemotherapy (*p* = 0.597). Detailed survival outcomes by treatment are described in Table 2.

In multivariate analysis, patients who received NCRT+S had a significantly reduced risk of death compared with patients who underwent upfront surgery alone (HR 0.77; 95% CI, 0.64–0.93; *p* = 0.007). Moreover, surgery with adjuvant chemotherapy was also associated with a significantly reduced risk of death in comparison with surgery alone (HR 0.75; 95% CI, 0.61–0.93; *p* = 0.009; Table 3). In particular, for patients with positive nodes after upfront surgery, adjuvant chemotherapy was shown to significantly reduce the risk of death (HR 0.65; 95% CI, 0.53–0.81; *p* < 0.001). The multivariate Cox proportional-hazard model stratified by stage and type of treatment is detailed in the Appendix A.

## 4. Discussion

Surgery is the standard treatment for ESCC; however, the associated survival rate is poor. Therefore, the treatment of ESCC has been evolving in conjunction with preoperative or postoperative therapies, including chemotherapy, radiotherapy, and combined therapy. For the past few decades, several prospective randomized controlled trials (RCTs) have sought to assess the efficacy of neoadjuvant or adjuvant therapies [7,8,9,10]. However, owing to the limitations of the therapeutic and diagnostic methods, the small sample sizes, and the relatively low quality of these studies, their results were insufficient to allow a conclusion about such efficacy. Recent advances in radiology, especially PET-CT, have made it possible to accurately detect systemic metastases; in addition, therapeutic methods including radiotherapy, surgery, and systemic therapies have been improved. All of these factors may lead to more high-quality RCTs, which can demonstrate the efficacy of NCRT. According to the CROSS trial and the NEOCRTEC5010 study, patients with ESCC who received NCRT followed by esophagectomy had significantly higher OS than those who received upfront surgery. Currently, neoadjuvant treatment followed by surgery is widely used for patients with locally advanced ESCC.

Although several RCTs have demonstrated the effectiveness of multimodal treatment for locally advanced ESCC, data shedding light on real-world treatment patterns in uncontrolled patients with locally advanced ESCC are currently lacking. Clinical trials are conducted in experimental settings with restricted patient populations with a good performance status. Meanwhile, RWE studies are largely conducted in real-world settings with a wide variety of patients, including frail patients and those with comorbidities who are not typically included in conventional clinical trials. RWE can be used to investigate whether multimodal treatment is actually effective in diverse patients in real-world settings. In addition, RWE represents trends and different patterns of treatment within the real-world population over time.

In this study, we aimed to analyze RWE to better understand treatment patterns and trends in outcomes for a real-world population with locally advanced ESCC. In our RWE study of 2151 patients, we found a significant improvement in survival outcome for locally advanced ESCC over time. Before the introduction of PET-CT (1994–2002), the 5-year OS of patients treated for locally advanced ESCC was 36.8% (31.4–43.1%). However, after the use of PET-CT evaluation for cancer staging became routine (2003–2012), the 5-year OS was improved to 42.2% (38.8–45.7%). Notably, since 2013, when the use of NCRT for advanced ESCC became routine, the 5-year OS has greatly improved, reaching 56.3% (53.2–59.6%). Regarding treatment modality, NCRT+S was proven to be the most effective treatment for locally advanced ESCC. In addition, NCRT+S significantly improved survival (58.1% (53–63.7%)), compared with upfront surgery (48.6% (45.9–51.5%; *p* < 0.001)). This finding broadly supports the current treatment guideline for locally advanced ESCC, which recommend the NCRT followed by surgery.

Combined chemotherapy and radiotherapy in the neoadjuvant setting has several advantages: chemotherapy can control distant metastatic tumor cells and radiotherapy can control local tumors; thus, spatial cooperation may enhance antitumor efficacy. Therefore, most guidelines recommend NCRT before surgery for patients with locally advanced ESCC [11,12]. Since the CROSS trial was published, we have routinely used NCRT before surgery for locally advanced ESCC. At our institution to date, 419 patients have received NCRT+S and 1299 have undergone upfront surgery. In our current RWE study, we comprehensively analyzed the clinical trend of locally advanced ESCC patients over a 24-year period, including baseline characteristics, the patterns of treatment and outcomes, and sequential treatments. The results from this study suggest that patients with locally advanced ESCC may benefit more from neoadjuvant CRT, which is in accordance with previous findings in the literature.

Regarding postoperative treatment, adjuvant chemotherapy may provide a survival benefit for patients with positive lymph nodes or locally advanced ESCC in clinical practice. However, this benefit has not been evaluated in a phase III RCT. The reason is that most patients who undergo esophagectomy have a poor performance status after surgery and experience postoperative complications and discomfort, including postoperative pain, poor oral intake, and fatigue. The 2019 National Comprehensive Cancer Network guidelines for the diagnosis and treatment of esophageal cancer and gastroesophageal junction carcinoma recommend that, regardless of pT or pN staging, no additional treatment other than surveillance is needed for patients who have undergone R0 resection. In contrast, the latest Japanese Esophageal Society guidelines recommend postoperative chemotherapy for clinical stage II or III esophageal cancer patients who have undergone surgery without preoperative therapy, although only weak evidence supports these guidelines. According to several studies, adjuvant chemotherapy may provide a survival benefit for patients with positive lymph nodes or patients with relatively advanced stages in clinical practice [7,13,14,15,16,17,18,19,20,21,22]. However, this benefit has rarely been investigated in RCTs. In the present study, upfront surgery with adjuvant chemotherapy significantly reduced the risk of death in comparison with surgery alone (HR 0.69; 95% CI, 0.56–0.85). Surprisingly, patients who received upfront surgery followed by adjuvant chemotherapy showed comparable outcomes to patients with NCRT+S in the practice setting (58.4% (52.7–64.7%) vs. 58.1% (53–63.7%)). It seems possible that patients who could tolerate adjuvant chemotherapy after surgery were highly selected with good performance. In practice, only 22% of patients with upfront surgery received adjuvant chemotherapy or CCRT in this study. Most patients could not receive chemotherapy because of medically compromising conditions, including old age, poor general condition, underlying comorbidities, postoperative complications/mortality, or early relapse of cancer. Therefore, these findings cannot be extrapolated to all patients. Additionally, we found that adjuvant chemotherapy was shown to significantly reduce the risk of death, especially in patients with positive nodes after upfront surgery (HR 0.65; 95% CI, 0.53–0.81). It can thus be suggested that adjuvant chemotherapy may be helpful for selected patients who have undergone upfront surgery with positive nodes.

Recently, a phase III double-blind RCT trial (CheckMate 577) demonstrated that a checkpoint inhibitor as adjuvant therapy improved disease-free survival in patients with resected esophageal cancer who had received neoadjuvant CRT [23]. Studies of the benefit of adjuvant checkpoint inhibitor therapy in patients undergoing definitive chemoradiotherapy are ongoing [24]. The checkpoint inhibitor as adjuvant therapy is a new standard therapeutic option for locally advanced ESCC.

This study has several limitations. First, in addition to the retrospective nature of this study, this investigation involved only a single-center study, which makes it difficult to generalize or extrapolate to other settings. Second, one feature of this cohort is that it included a large number of surgical cases. Our institution is an academic tertiary hospital at which 2151 patients were treated for locally advanced ESCC. Improvement in quality and process in surgery during the study period might affect survival; therefore, survival outcomes according to treatment period need to be interpreted with caution. Third, we analyzed only ESCC patients due to a high prevalence of ESCC, so whether these results are applicable in Western countries with a high prevalence of esophageal adenocarcinoma warrants additional investigation. Fourth, information on toxicities or morbidities was not available. 

## 5. Conclusions

In conclusion, this analysis of real-world data demonstrated a significantly improved survival outcome for patients with locally advanced ESCC over time since NCRT before surgery has become routine. Our results showed that NCRT followed by surgery was the most effective treatment for locally advanced ESCC and that adjuvant chemotherapy may be an encouraging therapeutic option for patients with positive nodes after upfront surgery.

## Figures and Tables

**Figure 1 cancers-14-04953-f001:**
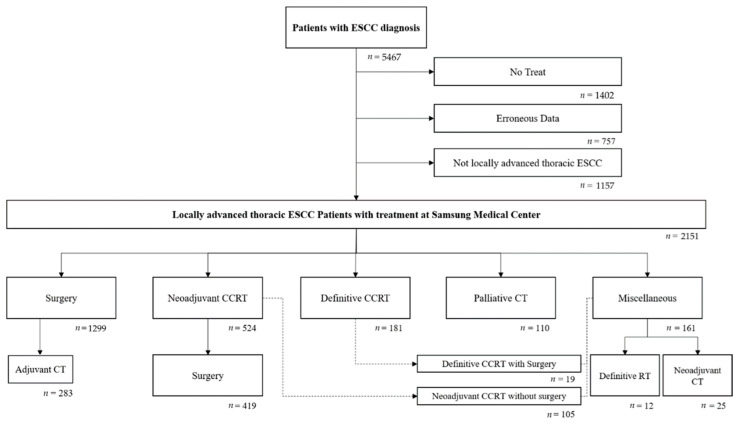
Eligibility process.

**Figure 2 cancers-14-04953-f002:**
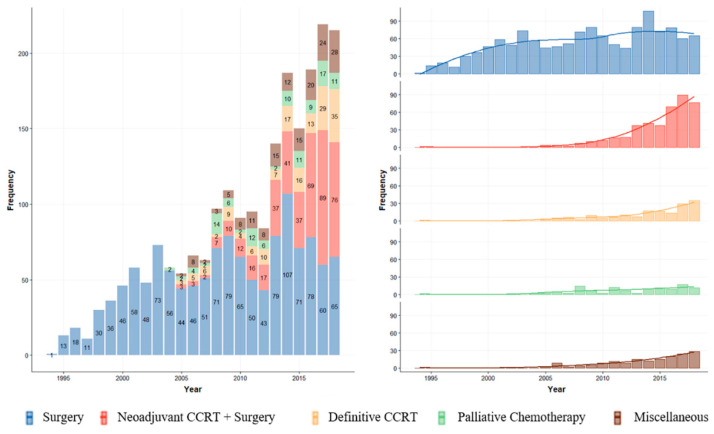
Annual trend of treatment for locally advanced ESCC.

**Figure 3 cancers-14-04953-f003:**
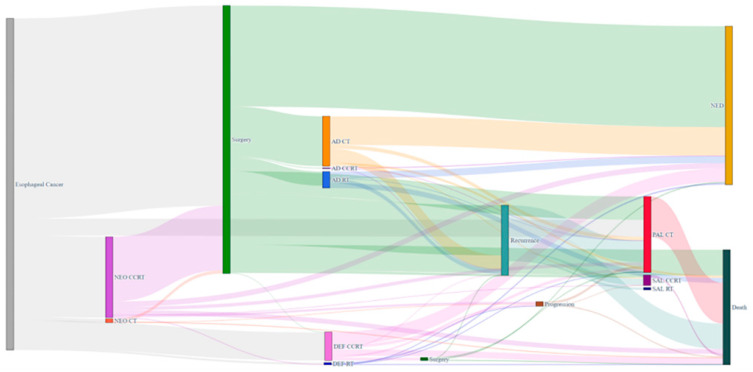
Sankey diagram for visualization of treatment and outcome patterns for 2 years after diagnosis of patients with locally advanced esophageal cancer. The thickness of the line in the figure shows how many patients belonged to each specific treatment.

**Figure 4 cancers-14-04953-f004:**
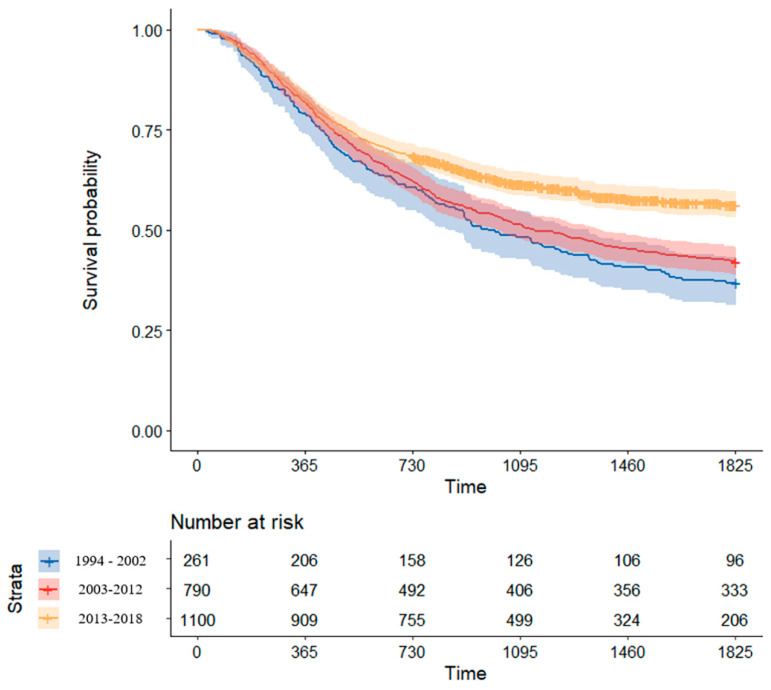
Comparison of Kaplan–Meier survival curve by treatment period.

**Figure 5 cancers-14-04953-f005:**
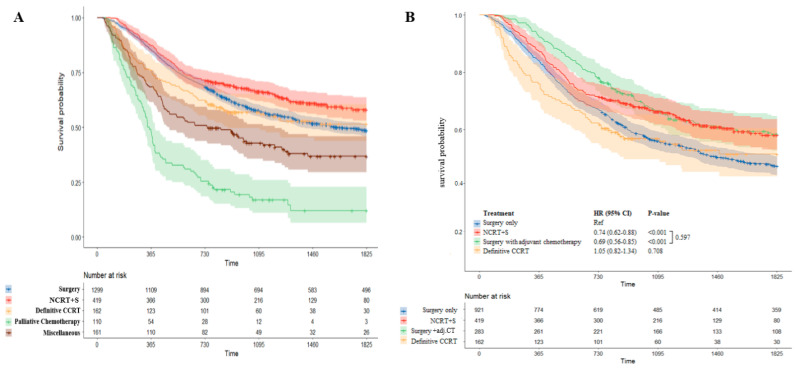
Comparison of Kaplan–Meier survival curve by treatment modality. Survival outcomes according to the type of initial treatment are shown in (**A**). Surgery group was divided into patients with adjuvant chemotherapy and without adjuvant chemotherapy. A comparison of survival curves between these subgroups is presented in (**B**).

**Table 1 cancers-14-04953-t001:** Characteristics of participants.

	Total (*n* = 2151)	Surgery (*n* = 1299)	NCRT + Surgery (*n* = 419)	Definitive CCRT (*n* = 162)	Palliative CT (*n* = 110)	Miscellaneous (*n* = 161)	*p*-Value
Age, (mean ± sd)	63.1 ± 8.5	63.3 ± 8.4	62.7 ± 8.3	61.8 ± 9.2	62.7 ± 8.4	64.0 ± 9.4	0.095
Sex, *n* (%)							0.074
Female	129 (6.0%)	67 (5.2%)	30 (7.2%)	17 (10.5%)	6 (5.5%)	9 (5.6%)	
Male	2022 (94.0%)	1232 (94.8%)	389 (92.8%)	145 (89.5%)	104 (94.5%)	152 (94.4%)	
ECOG PS, *n* (%)							<0.001
0	1637 (76.1%)	976 (75.1%)	382 (91.2%)	104 (64.2%)	71 (64.5%)	104 (64.6%)	
1	180 (8.4%)	77 (5.9%)	29 (6.9%)	28 (17.3%)	25 (22.7%)	21 (13.0%)	
≥2	57 (2.6%)	26 (2.0%)	8 (1.9%)	9 (5.6%)	7 (6.4%)	7 (4.3%)	
Unknown	277 (12.9%)	220 (16.9%)	0 (0.0%)	21 (13.0%)	7 (6.4%)	29 (18.0%)	
Smoking, *n* (%)							<0.001
Never smoker	1075 (50.0%)	742 (57.1%)	103 (24.6%)	87 (53.7%)	60 (54.5%)	83 (51.6%)	
Ex-smoker	998 (46.4%)	515 (39.6%)	305 (72.8%)	68 (42.0%)	45 (40.9%)	65 (40.4%)	
Current smoker	78 (3.6%)	42 (3.2%)	11 (2.6%)	7 (4.3%)	5 (4.5%)	13 (8.1%)	
Comorbidity, *n* (%)							
Hypertension	611 (28.4%)	352 (27.1%)	147 (35.1%)	39 (24.1%)	35 (31.8%)	38 (23.6%)	0.007
Diabetes mellitus	252 (11.7%)	154 (11.9%)	57 (13.6%)	18 (11.1%)	13 (11.8%)	10 (6.2%)	0.182
Tuberculosis	123 (5.7%)	81 (6.2%)	19 (4.5%)	9 (5.6%)	5 (4.5%)	9 (5.6%)	0.731
Hepatitis	54 (2.5%)	33 (2.5%)	10 (2.4%)	6 (3.7%)	0 (0.0%)	5 (3.1%)	0.401
Stage, *n* (%)							
T							<0.001
0	30 (1.4%)	2 (0.2%)	23 (5.5%)	1 (0.6%)	1 (0.9%)	3 (1.9%)	
1	371 (17.2%)	275 (21.2%)	46 (11.0%)	14 (8.6%)	12 (10.9%)	24 (14.9%)	
2	311 (14.5%)	178 (13.7%)	69 (16.5%)	19 (11.7%)	16 (14.5%)	29 (18.0%)	
3	1286 (59.8%)	761 (58.6%)	266 (63.5%)	95 (58.6%)	68 (61.8%)	96 (59.6%)	
4	153 (7.1%)	83 (6.4%)	15 (3.6%)	33 (20.4%)	13 (11.8%)	9 (5.6%)	
N							
0	377 (17.5%)	226 (17.4%)	78 (18.6%)	36 (22.2%)	15 (13.6%)	22 (13.7%)	<0.001
1	1266 (58.9%)	843 (64.9%)	217 (51.8%)	76 (46.9%)	49 (44.5%)	81 (50.3%)	
2	386 (17.9%)	170 (13.1%)	105 (25.1%)	39 (24.1%)	24 (21.8%)	48 (29.8%)	
3	122 (5.7%)	60 (4.6%)	19 (4.5%)	11 (6.8%)	22 (20.0%)	10 (6.2%)	
Location of tumor, *n* (%)							<0.001
Cervical	51 (2.4%)	9 (0.7%)	2 (0.5%)	29 (17.9%)	4 (3.6%)	7 (4.3%)	
Thoracic	1939 (90.1%)	1198 (92.2%)	403 (96.2%)	115 (71.0%)	89 (80.9%)	134 (83.2%)	
Abdominal	16 (0.7%)	11 (0.8%)	0 (0.0%)	0 (0.0%)	2 (1.8%)	3 (1.9%)	
NOS	106 (4.9%)	66 (5.1%)	9 (2.1%)	9 (5.6%)	12 (10.9%)	10 (6.2%)	
Overlapping lesion	39 (1.8%)	15 (1.2%)	5 (1.2%)	9 (5.6%)	3 (2.7%)	7 (4.3%)	

NCRT, neoadjuvant chemoradiotherapy; CCRT, concurrent chemoradiotherapy; CT, chemotherapy; ECOG, European Cooperative Oncology Group; PS, performance status; NOS, not otherwise specified.3.3. Treatment Trends and Pattern.

**Table 2 cancers-14-04953-t002:** Five-year survival rate according to treatment.

Initial Treatment	Subsequent Treatment	Five-Year Survival Rate (95% Confidence Intervals)		
		RFS	PFS	OS
Surgery	Overall (*n* = 1299)	0.409 (0.382–0.437)		0.486 (0.459–0.515)
	None (*n* = 921)	0.393 (0.361–0.427)		0.473 (0.441–0.507)
	Adjuvant CT (*n* = 283)	0.500 (0.443–0.565)		0.584 (0.527–0.647)
	Adjuvant RT (*n* = 90)	0.296 (0.214–0.410)		0.330 (0.245–0.444)
	Adjuvant CCRT (*n* = 5)	0.400 (0.137–1.000)		0.533 (0.214–1.000)
Neoadjuvant CCRT	Overall (*n* = 524)	0.341 (0.299–0.389)		0.532 (0.487–0.582)
	Surgery (*n* = 419)	0.365 (0.318–0.419)		0.581 (0.530–0.637)
	None (*n* = 380)	0.343 (0.293–0.400)		0.572 (0.518–0.633)
	Adjuvant RT (*n* = 1)	1.000 (1.000–1.000)		1.000 (1.000–1.000)
	Adjuvant CT (*n* = 38)	0.549 (0.411–0.734)		0.624 (0.485–0.802)
	None (*n* = 103)		0.241 (0.161–0.359)	0.332 (0.247–0.445)
	Definitive RT (*n* = 2)		1.000 (1.000–1.000)	1.000 (1.000–1.000)
Definitive RT	Overall (*n* = 12)		0.643 (0.412–1.000)	0.643 (0.412–1.000)
Neoadjuvant CT	Overall (*n* = 25)	0.200 (0.080–0.502)		0.223 (0.088–0.229)
	None (*n* = 4)		1.000 (1.000–1.000)	1.000 (1.000–1.000)
	Surgery (*n* = 21)	0.244 (0.100–0.596)		0.279 (0.144–0.680)
Definitive CCRT	Overall (*n* = 181)	0.427 (0.357–0.512)		0.517 (0.444–0.601)
	None (*n* = 162)	0.436 (0.361–0.527)		0.515 (0.438–0.606)
	Surgery (*n* = 19)	0.361 (0.197–0.663)		0.526 (0.344–0.806)
Palliative CT	Overall (*n* = 110)			0.121 (0.064–0.229)

**Table 3 cancers-14-04953-t003:** Hazard ratio estimated in univariate and multivariate Cox proportional-hazard models.

	Univariate Analysis		Multivariate Analysis	
Variables	Hazard Ratio (95% CI)	*p*-Value	Hazard Ratio (95% CI)	*p*-Value
Treatment				
Surgery only	Ref		Ref	
Neoadjuvant CCRT with surgery	0.737 (0.615–0.882)	<0.001	0.770 (0.636–0.931)	0.007
Surgery with adjuvant chemotherapy	0.690 (0.561–0.848)	<0.001	0.751 (0.605–0.931)	0.009
Definitive CCRT	1.048 (0.820–1.339)	0.7076	0.852 (0.661–1.099)	0.217
Palliative CT	3.384 (2.700–4.241)	<0.001	1.889 (1.454–2.454)	<0.001
Miscellaneous	1.481 (1.237–1.774)	<0.001	1.267 (1.053–1.524)	0.012
Age	1.009 (1.001–1.017)	0.033	1.010 (1.003–1.017)	0.009
Sex				
Female	Ref		Ref	
Male	1.712 (1.232–2.379)	0.001	1.628 (1.194–2.222)	0.0021
ECOG PS				
0	Ref		Ref	
1	1.658 (1.336–2.057)	<0.001	1.236 (1.006–1.518)	0.044
≥2	2.279 (1.628–3.188)	<0.001	1.638 (1.190–2.256)	0.003
Unknown	1.723 (1.433–2.072)	<0.001	1.462 (1.227–1.741)	<0.001
Smoking			Not significant	
Current smoker	Ref			
Ex-smoker	0.848 (0.585–1.229)	0.383		
Never smoker	1.109 (0.767–1.603)	0.583		
Hypertension			Not significant	
No	Ref			
Yes	0.970 (0.839–1.121)	0.681		
Diabetes mellitus			Not significant	
No	Ref			
Yes	1.145 (0.945–1.387)	0.167		
Tuberculosis			Not significant	
No	Ref			
Yes	1.035 (0.790–1.357)	0.801		
Hepatitis				
No	Ref		Ref	
Yes	1.548 (1.076–2.227)	0.019	1.819 (1.292–2.561)	<0.001
Cancer Stage				
T stage				
T0	Ref		Ref	
T1	1.286 (0.6–2.756)	0.518	0.867 (0.398–1.891)	0.720
T2	1.846 (0.863–3.949)	0.114	1.119 (0.516–2.427)	0.776
T3	2.768 (1.315–5.827)	0.007	1.709 (0.802–3.644)	0.165
T4	4.63 (2.155–9.948)	<0.001	2.552 (1.169–5.574)	0.019
N stage				
N0	Ref		Ref	
N1	1.612 (1.34–1.938)	<0.001	1.716 (1.420–2.075)	<0.001
N2	1.743 (1.403–2.166)	<0.001	1.889 (1.507–2.368)	<0.001
N3	3.079 (2.349–4.035)	<0.001	2.570 (1.931–3.421)	<0.001
M stage				
M0	Ref		Ref	
M1	2.754 (2.348–3.231)	<0.001	1.937 (1.617–2.320)	<0.001

## Data Availability

The data presented in this study are available on request from the corresponding author. The data are not publicly available due to privacy restrictions.

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
