# Peer review of "Clinical Trends in Management of Locally Advanced ESCC: Real-World Evidence from a Large Single-Center Cohort Study"

_cancers, 2022, doi:10.3390/cancers14194953_

Round 1

Reviewer 1 Report

Neadjuvant chemoradiation (nCRT) is the main stay of treatment for ESCC. In this large retrospective multi center cohort study they derive secondary evidence for the advantage of neo-adjuvant chemoradiation above surgery by comparing previous practice until with recent practice combining the modalities, and describe changes in time of treatment of locoregional ESCC. For this they use their large academic hospital data with over 2000 patients treated. After introduction of neadjuvant CRT 5-year survival increased up to 58% compared to 49% for upfront surgery. 

Major comments:

- The methods lack overview of given treatment e.g. which radiotherapy and oncological regimens were given. All treatment groups need to be described for clearly, why did some patients go for surgery after definitive CRT, salvage therapy? Why did some go for palliative treatment with locoregional disease, comorbidity? Why did some patients receive only me-adjuvant chemo or RT.

- The authors should perform an uni- and multivariate analysis to correct for confounding factors such as T and N stage

- The introduction states surgery is currently standard of care, but in many countries nCRT+surgery is the current standard of care as many previous RCT's have shown the benefit. Although I agree their large scale real life data adds to the existing data I would not imply we currently lack sufficient evidence for nCRT+surgery. This nuance is better described in the discussion and I would advise to specify this more clearly in the introduction.

Minor:

- is there any data available on the toxicity/morbidity of the patients?

- which statistical method was used for table 1?

- Inclusion criteria: should state ESCC instead of esophageal cancer

- fig 1: no treat should be no treatment

- table 1 'cancer of origin' should this read: location of tumor? somewhat confusion this origin term

Author Response

We thank you for your time and encouraging comments. To address your comments and suggestions, we revised the manuscript significantly. Please see the attachment

Reviewer 2 Report

The authors investigated their experience of locally advanced esophageal cancer as a real-world experience (RWE). This is a single center cohort study from a high-volume hospital in South Korea.

・Based on CheckMate 577 study, NCRT+S followed by Nivolumab is a standard therapy for locally advanced esophageal cancer. (N Eng J Med. 2021 Apr;384(13):1191-1203) The authors should refer and discuss about this.

・Patients with NCRT+S followed by adjuvant therapy showed better survival, but how was the therapy (i.e. surgery alone, NCRT+S, definitive CCRT, and palliative CT) chosen? The authors descried patients with worse PS or unresectable tumor underwent definitive CCRT, but I guess patients with cervical esophageal cancer seems to undergo definitive CCRT in Table 1. Additionally, what kind of patients underwent palliative CT?  Why patients with upfront surgery did not undergo NCRT?  The authors should mention about this point in detail.

・The conclusion that neoadjuvant CCRT improved survival is not a new finding, even if this is a RWE. What is the difference between this RWE and previous RCT? If the patient’s characteristics were different, the authors should describe and discuss about this point clearly.

・From 1994, surgical procedure must be changed, from open surgery to laparoscopic/thoracoscopic surgery. This trend may also affect postoperative survival. The authors should mention about this point.

・With NCRT, did complete resection by surgery increase? This may also lead to better survival in NCRT group. The authors should show the data about this.

・Detailed information about NCRT or CT including radiotherapy dose and anticancer agent should be shown.

Author Response

(The authors gave the same response as above.)

Round 2

Reviewer 1 Report

Thank you for making the adjustments to the paper. There are several items which however still need improvement

1). It is a very heterogeneous cohort and although the M&M now provides more information on the given treatment, it still lacks information about the given adjuvant CT and RT. This is relevant because when comparing the RFS and OS in table 2 between surgery only without adjuvant treatment and nCRT they are nearly equivalent 39 vs 36%.  Is there an effect of the adjuvant treatment? have the authors corrected for this? and can they provide information about the adjuvant schemes

2) the added paragraphs contain many grammatical errors and this needs improvement

3) Why did the authors choose not to meet the proportional hazards? As stated in the M&M not assuming these. In table 3 I still can't see a correction for the TNM stage

Author Response

We thank you for your time and encouraging comments. To address your comments and suggestions, we revised the manuscript significantly. Please see the attachment for our specific responses as well as our modifications to the manuscript.

Reviewer 2 Report

The manuscript was properly revised. I think this manuscript deserves publication.

Author Response

We thank you for your time and encouraging comments.

Round 3

Reviewer 1 Report

Thank you for the substantial improvements, I would advise one last grammar check as there are still grammatical errors in the document